# Milling-Assisted Loading of Drugs into Mesoporous Silica Carriers: A Green and Simple Method for Obtaining Tunable Customized Drug Delivery

**DOI:** 10.3390/pharmaceutics15020390

**Published:** 2023-01-24

**Authors:** Basma Moutamenni, Nicolas Tabary, Alexandre Mussi, Jeremy Dhainaut, Carmen Ciotonea, Alexandre Fadel, Laurent Paccou, Jean-Philippe Dacquin, Yannick Guinet, Alain Hédoux

**Affiliations:** 1UMR 8207, UMET—Unité Matériaux et Transformations, University Lille, CNRS, INRAE, Centrale Lille, F-59000 Lille, France; 2UMR 8181, UCCS—Unité de Catalyse et Chimie du Solide, University Lille, CNRS, Centrale Lille, University Artois, F-59000 Lille, France; 3Unité de Chimie Environnementale et Interactions sur le Vivant—UCEIV, UR4492, Université du Littoral Côte d’Opale, F-59140 Dunkerque, France

**Keywords:** milling, physical state, low-frequency Raman spectroscopy, scanning electron microscopy, drug release profiles

## Abstract

Mesoporous silica (MPS) carriers are considered as a promising strategy to increase the solubility of poorly soluble drugs and to stabilize the amorphous drug delivery system. The development by the authors of a solvent-free method (milling-assisted loading, MAL) made it possible to manipulate the physical state of the drug within the pores. The present study focuses on the effects of the milling intensity and the pore architecture (chemical surface) on the physical state of the confined drug and its release profile. Ibuprofen (IBP) and SBA-15 were used as the model drug and the MPS carrier, respectively. It was found that decreasing the milling intensity promotes nanocrystallization of confined IBP. Scanning electron microscopy and low-frequency Raman spectroscopy investigations converged into a bimodal description of the size distribution of particles, by decreasing the milling intensity. The chemical modification of the pore surface with 3-aminopropyltriethoxisylane also significantly promoted nanocrystallization, regardless of the milling intensity. Combined analyses of drug release profiles obtained on composites prepared from unmodified and modified SBA-15 with various milling intensities showed that the particle size of composites has the greatest influence on the drug release profile. Tuning drug concentration, milling intensity, and chemical surface make it possible to easily customize drug delivery.

## 1. Introduction

Nowadays, the discovery of active pharmaceutical ingredients (APIs) with high therapeutic value exhibits a significant increase. However, most of the new drug candidates, which are in the pipelines of the pharmaceutical industry, are synthesized in the crystalline state and are often poorly water soluble, with unavoidable consequences on bioavailability [1]. Among a variety of approaches to improve solubility, dissolution rates, and bioavailability, the use of mesoporous silica (MPS) matrices as drug carriers are now considered as promising drug delivery systems [2,3,4,5,6] because of their high surface area, tunable pore diameter and morphology, and easily modifiable chemical surface. Additionally, their inorganic nature makes them a protective shield for active molecules against physiological attacks. It is recognized that confinement in MPS is a pathway for manipulating the physical state of APIs, making it possible to stabilize active molecules in nanocrystalline or amorphous states by varying the pore diameter [7,8]. The unstable nature of amorphous materials, considered as a substantial drawback for the use of amorphous APIs to formulate oral dosage forms, can be overcome if the pore size is smaller than the critical radius of nucleation, inhibiting the nucleation and growth process. Unfortunately, their weak loading capacity and the toxicity of organic solvents used for loading APIs in solution were considered insurmountable obstacles to their development [9]. As a consequence, very close attention has been focused on loading methods [5,10]. They are classified into two categories, namely, the solvent-based and solvent-free methods. The solvent-based methods such as adsorption, incipient wetness impregnation, solvent evaporation, spray-drying, etc., are the most commonly used despite the often-necessary use of toxic organic solvents that are incompatible with the therapeutic application. Solvent-free methods show better compatibility with the therapeutic domain but are used much less. The best-known method in this category is the melt method, which is considered very effective [11] but can only be used for APIs with low melting temperatures [12] since melting at high temperatures could be responsible for chemical degradation and loss of therapeutic activity. It was shown that it is possible to easily load APIs into MPS by physical mixing [13]. Based on this concept, an innovative method has been developed called the milling-assisted loading (MAL) method [14], consisting of co-milling API and MPS. Co-milling the MPS and API for 30 min using an oscillating mixer at 30 Hz makes it possible to load a large amount of API (35 to 40 wt%) while preserving the ordered structure of the MPS [14,15], without the use of any organic solvent. This green method also allows the loading of a large amount of hydrophobic molecular materials, as recently shown for L-tryptophan [16], while only a very weak amount can be loaded using the most classic solvent-based methods [17,18].

The originality of the MAL method is the loading of solid-state materials that make crystallization in very small pore diameters (∅ < 7 nm) possible from pre-existing nuclei bypassing the nucleation process [19]; this is not possible using solvent-based methods. This singularity gives the opportunity to explore new physical states [20] and atypical phase transformation driven by the instability of nanocrystals inherent to the pore geometry [19].

Although this method has already been the subject of several studies aiming at optimizing the operating parameters (milling duration and filling capacity), this area is still in its infancy, and some additional parameters require investigation. The aim of this study was firstly focused on analyzing the influence of milling frequency on the material loaded into the MPS (physical state, filling degree, and dissolution rate). Secondly, the same characteristics of the loaded material were analyzed after changing the chemical surface.

SBA-15 was selected as the model MPS and racemic ibuprofen (IBP) as the model drug. Selecting SBA-15 allows tuning the pore diameter without changing the pore geometry. The previous analyses performed on the confinement of IBP within SBA-15 matrices of different pore sizes were used as references. The choice of IBP was inherent to its T_g_ value (about −50 °C), making its amorphization by milling impossible at room temperature [21]. Consequently, the absence of crystallized IBP after co-milling directly indicates that IBP is confined in SBA-15.

Most of the investigations of the physical state of IBP in the IBP:SBA-15 composites were performed using Raman spectroscopy. Combining low- and mid-frequency Raman investigations provides accurate information on the physical state of IBP (amorphous, nano/micro-crystalline), on the molecular conformation of IBP, and on the interaction between the pore surface and IBP via H-bonding. Previous studies have shown that the main advantage of this technique, compared with X-ray diffraction, is that the Raman signal of the MPS matrix is insignificant compared with that of APIs including IBP. Scanning electron microscopy experiments were performed to analyze the influence of milling on SBA-15.

## 2. Materials and Methods

### 2.1. Materials

Racemic ibuprofen (designated herein as IBP) was purchased from Sigma, Saint Louis, MO, USA, (CAS number 15687-27-1, purity ≥ 99.8% GC assay) and was used without further purification. SBA-15 purchased from Sigma Aldrich was used as the MPS. SBA-15 carriers were composed of ordered cylindrical channels characterized by a 2D hexagonal symmetry, and the channel size was determined in the present study by nitrogen physisorption measurements.

### 2.2. Methods

#### 2.2.1. Milling-Assisted Loading (MAL) Method

Ibuprofen was loaded using the MAL method. Co-milling of IBP and SBA-15 solid mixtures was systematically performed at room temperature using a Retsch Mixer Mill MM400 (Haan, Germany). About 400 mg of as-received crystalline IBP and MPS powders were placed in an Eppendorf container for milling using an oscillating milling device (MM400—Retsch) for 30 min and one stainless steel ball (Ø = 7 mm) at frequencies between 10 Hz and 30 Hz. To avoid any overheating of the IBP, milling periods (5 min) were alternated with pause periods (2 min). The influence of milling on the physical state of the IBP was analyzed in Figure A1 in Appendix A. Spectra of the IBP (as-received, milled at 10 Hz and 30 Hz) are plotted in Figure A1. The comparison shows a slight broadening of phonon peaks corresponding to the size reduction in crystallites, which is mostly significant for the sample milled at 30 Hz, for which melting starts at a lower temperature than for the two other samples. Several composites were prepared corresponding to co-milling of the as-received Ibuprofen with the SBA-15 matrix in an IBP weight % of X = 35 to ensure the complete loading of IBP [14]. The composites are hereinafter identified as IBPX%:SBA-15, with X being the weight % of IBP.

#### 2.2.2. Synthesis of Modified SBA-15 (SBA-15-NH_2_)

Chemical modification of the pore walls of SBA-15 was realized with APTES (3-aminopropyltriethoxisylane, H_2_N(CH_2_)_3_Si(OC_2_H_5_)_3_). In a typical procedure, 1 g of commercial SBA-15 was refluxed at 80 °C during 24 h in a 50 mL ethanolic solution containing 1 mL APTES under continuous stirring. The resulting sample was filtered, washed repeatedly with ethanol, and then dried at 70 °C for 12 h. The obtained sample was designated as SBA-15-NH_2_.

#### 2.2.3. Characterization of Mesoporous Silica Carriers

Nitrogen physisorption measurements were carried out at 77 K using a Micromeritics Tristar II Plus apparatus. Prior to the measurements, the samples were outgassed at 423 K for 8 h in the case of unloaded matrices, and at room temperature for 24 h in the case of loaded matrices. The specific surface areas were calculated from the linear portion of the Brunnauer–Emmett–Teller (BET) plots, and the pore size distributions were evaluated by the Barrett–Joyner–Halenda (BJH) method from the adsorption branch of nitrogen isotherms. The total pore volume was determined at p/p_0_ = 0.95.

#### 2.2.4. Electron Microscopy

Transmission electronic microscopy (TEM) investigations were performed with a FEI^®^ Tecnai G^2^20Twin microscope (Plateforme de Microscopie Electronique de Lille, University of Lille, Lille, France), operating at 200 kV with a LaB_6_ filament, using a double tilt sample holder. The milled and un-milled SBA-15 powder specimens were applied on lacey carbon support TEM films.

For scanning electron microscopy (SEM) observations, the samples were deposited on lacey carbon support TEM films and installed in the SEM with a TEM grid holder. SEM images were recorded using the lower second electron detector present in the chamber of a FEG-SEM JEOL JSM-7800F LV (Plateforme de Microscopie Electronique de Lille, University of Lille, Lille, France). To reduce beam damage on samples, the conditions of observation were 5 kV accelerating voltage, smallest objective lens aperture (position 4), and 10 mm working distance.

#### 2.2.5. Drug Release Studies

The release of ibuprofen from the mesoporous matrices was monitored using an UV spectrometer (UV-1800 Spectrophotometer Shimadzu) by measuring the variations in absorbance at the wavelength of 220 nm, which correspond to the maximum of absorbance of ibuprofen in 0.1 M HCl solution.

Experimentally, at room temperature, 10 mg of composites (IBP:SBA-15/IBP:SBA-15-NH_2_) were added under stirring at 100 rpm to 250 mL of 0.1 M hydrochloric acid (pH = 1.1) respecting sink conditions [22,23]. For each measurement, 2 ml of solution was collected for analysis; then, to avoid volume variations, the solution was returned into the medium. The release experiments for each sample were replicated 3 times and the presented experimental points (taken every 2 min in the first moments of the experiment, and every 5 min when equilibrium was reached), which correspond to the amount of ibuprofen delivered as a function of time, are the mean with the standard deviation. The ratio of the absorbance at a time *t* over the absorbance at the time *t_∞_* defines the ibuprofen release percentage. Each drug release was monitored for 80 min.

#### 2.2.6. Low-Frequency Raman Spectroscopy

Low-frequency Raman spectroscopy investigations were carried out on the high-dispersive XY-Dilor spectrometer composed of three gratings (1800 gr/mm) and equipped with a Cobolt laser emitting at 660 nm. Maintaining the slits opened at 150 µm makes it possible to detect a Raman signal down to 5 cm^−1^ in a high-resolution configuration (lower than 1 cm^−1^). Composites were loaded in spherical Pyrex cells hermetically sealed. The temperature of the sample was regulated using an Oxford nitrogen flux device that keeps temperature fluctuations within 0.1 °C. Low-frequency Raman spectra (LFRS) were collected between 5 and 350 cm^−1^ in 1 min, in situ during the heating ramp at 1 °C/min. The analysis of the LFRS requires a specific processing [24,25] inherent to the spectrum distortion induced by the temperature via the Bose factor, mostly important at very low frequencies. To obtain the low-frequency spectrum independent of temperature fluctuations, the Raman intensity $I_{Raman}\left(\omega ,T\right)$ is converted into reduced intensity $I_{r}\left(\omega \right)$ via:(1)$$I_{r}\left(\omega \right)=\frac{I_{Raman}\left(\omega ,T\right)}{\left[n\left(\omega ,T\right)+1\right]\omega }$$
where n(ω,T) is the Bose factor. This representation of the LFRS is generally used for highlighting a molecular disorder corresponding to fast molecular motions, which are thermally activated and detected in the low-frequency region (5–50 cm^−1^) and considered as relaxational motions giving a contribution to the LFRS called quasielastic intensity (I_QES_). In disordered molecular systems, the structural information is contained into the pure vibrational spectrum. It is obtained by removing the contribution of the quasielastic intensity from the I_r_(ω)-spectrum which is converted into Raman susceptibility according to [26,27]:(2)$$\chi \prime \prime \left(\omega \right)=\omega \cdot I_{r}\left(\omega \right)=\frac{C\left(\omega \right)}{\omega }G\left(\omega \right)$$
where C(ω) is the coupling coefficient between light and vibration and G(ω) is the vibrational density of states (VDOS). χ″(ω) is recognized to be a representation very close to the VDOS [28].

## 3. Results

### 3.1. SBA-15, SBA-15-NH_2_ Characterization

According to the IUPAC classification [29], the adsorption–desorption isotherms of SBA-15 and SBA-15-NH_2_, plotted in Figure 1a, are of type IV with H1 hysteresis loops, indicative of ordered mesoporous materials with some microporosity. SBA-15 is well-known for its arrangement of 2D hexagonal pores [30]. After functionalization, a strong decrease in adsorbed N_2_ is observed, which is related to the partial filling of pores by APTES. This is particularly marked for the micropore region (p/p_0_ < 0.2), as APTES may be able to clog some of the micropores. Hence, the resulting BET surface area decreases from 793 m²/g to 334 m²/g, while the total pore volume decreases from 0.80 cm^3^/g to 0.40 cm^3^/g for SBA-15 and SBA-15-NH_2_, respectively. Moreover, a shift of the hysteresis towards lower p/p_0_ is observed as a first hint of the decrease in the mesopore diameter. Figure 1b further shows the pore size distribution as evaluated by the BJH method. After APTES grafting, the average pore size decreases from 7.1 nm for SBA-15 to 5.7 nm for SBA-15-NH_2_.

The content of 3-aminopropyl groups grafted on the surface of SBA-15-NH_2_ was evaluated by CHN (carbon, hydrogen, and nitrogen) elemental analysis giving an elemental composition of 7.35 wt% C, 2.00 wt% H, and 1.58 wt% N. This result was confirmed by a thermogravimetric analysis presented in Figure A2 in Appendix B. Experiments were also performed on composites loaded within SBA-15 and SBA-15-NH_2_ by MAL at 10 Hz, for which nanocrystals were observed. The comparison of N_2_ adsorption isotherms obtained with non-loaded and loaded matrices are plotted in Figure 1c,d. The significant downshift of isotherms obtained with loaded composites with respect to those obtained with non-loaded matrices confirms that IBP was loaded inside the channels, in addition to the absence of IBP recrystallization after melting. It should be noticed that 35 wt% of IBP is not the maximum filling capacity previously determined (37 wt%) for similar mesoporous silica carriers [14], which can explain the detection of a residual amount of adsorbed N_2_ corresponding to a few channels remaining free.

### 3.2. Raman Spectroscopy

The low-frequency spectrum (LFRS) of the undercooled liquid bulk state of IBP was directly plotted in Figure 2a in its raw Raman intensity representation. The LFRS of SBA-15 taken with the same acquisition time was also plotted in the same figure to show that its signal is negligible compared with that of IBP. After correction of the Bose factor, the spectrum obtained in the reduced intensity representation was plotted in Figure 2b. There is no direct perceptible change between Raman intensity and reduced intensity representations because of the large contribution of the quasielastic scattering in the very low-frequency region, which dominates the spectrum as expected in the very disordered molecular systems, and flattens the vibrational spectrum. Figure 2b shows the fitting procedure used to determine the contribution of quasielastic intensity to the spectrum. The quasielastic scattering is usually well represented by a Lorentzian shape centered at ω = 0. After removing this contribution, the reduced intensity can be converted into Raman susceptibility according the relation (2). The spectra of amorphous and crystalline IBP in the bulk form are plotted in the Raman susceptibility representation in Figure 3a. These can be used as references for analyzing spectra collected on IBP35%:SBA-15 composites loaded with different milling frequencies. These spectra were plotted in Figure 3b. It is clearly observed that the low-frequency band shape of composites is highly dependent on the milling frequency, with the detection of broadened phonon peaks from and below 20 Hz. The spectrum of the composite loaded at the milling frequency of 10 Hz shows very distinguishable barely broadened phonon peaks of the bulk crystalline form indicating a relatively high degree of crystallization. An enhanced intensity can be clearly detected below 40 cm^−1^, marked with an arrow in Figure 3b, which is the signature of the amorphous IBP.

The fitting procedure described in Figure 2b provides information on the Lorentzian peak associated with the quasielastic scattering. The normalization procedure of the integrated intensity of the Lorentzian peak by the intensity of vibrational components that are almost temperature independent provides information distinctive to the quasielastic intensity. It is recognized that the temperature dependence I_QES_(T) is similar to that of the mean-square displacement <u^2^>(T) and inversely proportional to $log\eta \left(T\right)\;$ [31,32], which is closely involved in the phase transition mechanisms including the glass transition. It was previously shown that the glass transition temperature (T_g_) could be determined as corresponding to the change in the slope of I_QES_(T) [33,34]. The spectra collected during heating of glassy IBP in the bulk form ($\dot{T}=1\;\deg/\min$) are plotted in the reduced intensity representation in Figure 4a. This figure clearly shows the successive phase transitions of IBP, i.e., the glass transition at −50 °C, the crystallization of the metastable phase (PII) at 10 °C, the melting of PII at 20 °C, the crystallization of the stable form PI at 25 °C, and the melting of PI at 80 °C. The quasielastic intensity (I_QES_) can be calculated from the integrated intensity of the Lorentzian peak or by integrating the I_r_(ω)-spectra in the very low-frequency range, giving the temperature dependence plotted in Figure 4b. This plot recreates the phase transition sequence observed from the spectra in Figure 4a. This representation was adopted for analyzing the temperature dependence of the IBP loaded within SBA-15 by co-milling at 10 Hz and 30 Hz. LFR spectra collected during the first and second heating ramps of composites prepared by MAL at 10 Hz are plotted in Figure 5a,b, respectively.

The temperature dependences of the quasielastic intensity calculated from these spectra were plotted in Figure 6a. A quick look at Figure 5a reveals the presence of phonon peaks which are the signature of nanocrystals within the pore, while preparing the same composite by co-milling at 30 Hz leads to a totally amorphous IBP within the pores. A first heating of the composite from room temperature up to 90 °C was performed in order to melt the nanocrystals, as observed in Figure 5a and Figure 6a, directly from the beginning of the heating ramp up to 60 °C. This broad temperature range of melting reflects a broad distribution of nanocrystal size, and very low stability of the smallest nanocrystals. The composite was then re-cooled at −100 °C and spectra were collected during a second heat ramp and plotted in Figure 5b. No trace of phonon peaks was detected in all spectra collected during the second heating run, indicating that the nanocrystals were well located inside the pores. If nanocrystals are located outside the pores, recrystallization should be observed either on cooling or on the second heating. This method, which consists of analyzing the composites upon first heating, upon cooling down to the low temperatures, and upon a second heating, was previously used to determine the filling capacity related to the MAL method. The plot of I_QES_ during the first heating was slightly larger at low-temperature than I_QES_ calculated in spectra of bulk crystalline IBP (between 25 and 75 °C), indicating that nanocrystals are coexisting with amorphous IBP. The I_QES_(T) curve calculated in spectra collected during the second heating ramp shows a change in the slope, distinctive of T_g_, slightly above T_g_ in the bulk amorphous IBP. It is noticeable that this slope break is smoother than that detected upon heating glassy IBP in the bulk form. A comparison between I_QES_(T) curves obtained by analyzing the composites prepared by co-milling at 10 Hz and 30 Hz is shown in Figure 6b. The two curves are similar, with a slight temperature shift between the two curves associated with a slight shift of T_g_, both being above the T_g_ of the bulk glassy state.

Similar experiments were performed on composites prepared from modified SBA-15 (noted SBA-15-NH_2_) mesoporous silica loaded by the MAL method at 10 Hz and 30 Hz. LFRS collected at room temperature were plotted in Figure 7a. This figure highlights a significant degree of crystallization of IBP inside SBA-15-NH_2_ loaded at 10 Hz compared with the spectrum taken in the composite prepared with the as-received SBA-15 loaded at 10 Hz. For loading at 30 Hz, nanocrystals are also detected, while IBP was observed as amorphous in the as-received SBA-15. This clearly indicates that the change in the chemical surface promotes the crystallization of IBP. Increasing the milling time of MAL has no great influence on the degree of crystallinity of IBP within SBA-15-NH_2_. These features can be confirmed by the plot of I_QES_(T) curves (in Figure 7b) corresponding to the first heating of the composites. The quasielastic intensity of IBP at room temperature is higher in the as-received SBA-15 loaded by co-milling at 10 Hz and decreases in modified SBA-15-NH_2_, becoming close to the value obtained in the bulk crystal for the composite prepared by MAL at 10 Hz. In this composite, the I_QES_(T) curve (X) exhibits a clear 2-step increase, the first being observed around 50 °C and the second at 74 °C, i.e., very close to the melting temperature of the bulk. This feature indicates two types of particle sizes. By contrast, the curve corresponding to the first heating of IBP35%:SBA-15-NH_2_ shows an almost continuous melting from 50 to 74 °C. On the other hand, the temperature dependence of I_QES_ determined from the analysis of the second heating of the composite (similar to that corresponding to the composite prepared at 30 Hz and not plotted) shows a value of T_g_ lower than that determined in the unmodified SBA-15.

The degree of crystallinity of IBP within the various types of composites prepared by MAL at various frequencies was calculated using a method developed by the authors for analyzing the crystallization process of indomethacin [35] or the degree of transformation of phase I into phase II in caffeine [36]. This method has been applied to the determination of the crystallinity degree and is described in Appendix C (Figure A3). The degrees of crystallinity resulting from this spectrum processing, associated with Figure A4, are reported in Table 1.

### 3.3. Electron Microscopy

From Malfait et al. [14], the porosity size of the SBA powder specimen does not change with the milling process (an example of a TEM micrograph of un-milled particles is visible in Figure 8d). However, milling modifies the particle sizes.

The low accelerating voltage used for SEM analyses does not allow the application of an automatic method for the measurement of the particle sizes. Consequently, they were measured with a manual tracing method. This methodology is described in Figure 8 (example of an un-milled SBA powder specimen): particles are traced by hand (Figure 8b), then isolated from each other (Figure 8c) to estimate their proportion and size distribution (Figure 8d) using the ImageJ software. Another example of a bimodal size distribution measurement is described in Figure 9. The distribution of the particle sizes becomes bimodal (small particles vs. big particles) with milling. The higher the frequency, the lower the big particle size (they are progressively eroding), while the size of small particles remains almost constant and equal to approximately 110 nm (see Table 2). Furthermore, the higher the frequency, the higher the proportion of small particles at the expense of the proportion of big particles (see Table 2). Figure 9d shows the evolution of the distribution of particle sizes with milling frequency.

### 3.4. Study of Release Kinetic Profiles of Ibuprofen from Mesoporous Matrices

The influence of the functionalization of the mesoporous silica and the milling frequency of the composites on the release of ibuprofen have been analyzed using composites freshly prepared with 35% of IBP and 65% of unmodified and modified mesoporous silica carriers (SBA-15/SBA-15-NH_2_). Composites were prepared by MAL at 10 Hz and 30 Hz. The experiments were carried out in acidic medium (pH = 1.1) and sink conditions (see details in Section 2.2.5). Measurements were also carried out 24 h after the monitoring of the drug release to ensure that the total mass of IBP was released. The IBP amount released corresponded well to the total amount of ibuprofen loaded within the mesoporous matrix. This was verified using the ibuprofen calibration curve performed in a 0.1 M hydrochloric acid medium plotted in Appendix D, Figure A5. The drug release profiles from SBA-15 and SBA-15-NH_2_ carriers are plotted in Figure 10a,b, respectively

These figures show that drug release profiles from carriers prepared with SBA-15 and SBA-15-NH_2_ co-milled at 30 Hz can be qualitatively described as a 2-step process, i.e., (i) a very fast release observed up to 5 min, most probably due to IBP molecules located in the core of the channels, followed by (ii) a slow release attributed to the IBP molecules, which interact with the surface of the pores. This description can also be observed in Figure A6 in Appendix D. Such a behavior was already reported in the literature for confined ibuprofen [7,37,38]. However, it is worth noting that IBP is totally amorphous in unmodified SBA-15 while it is partially but significantly crystallized in SBA-15-NH_2_.

In view of Figure 10a,b, the main information emerging from this study is the preponderant effect of the milling frequency on the drug release. Indeed, Figure 10a,b show similar profiles for composites prepared by co-milling at 10 Hz, characterized by an enhanced sustained release compared with those obtained by co-milling at 30 Hz. This can be better visualized in Figure A6 in Appendix D. To illustrate this effect, the time required to release 80% of the IBP was studied (see Figure 10). For IBP35%:SBA-15 composites, 80% of drug release was reached after ~15 min in the case of the composite milled at 10 Hz, while it takes only ~5 min to achieve this release rate in the case of the composite milled at 30 Hz. This behavior can also be observed for IBP35%:SBA-15-NH_2_ composites. This effect can be explained by the size reduction in the porous particles, which may induce a reduction in the length of the channels of SBA-15 by increasing the frequency, hence, allowing a faster release. Composites prepared with modified or unmodified carriers by milling at 30 Hz exhibit very similar profiles despite the difference of crystallinity degree (see Table 1). This reveals the strong influence of the size distribution of mesoporous silica carriers on the drug release profiles. However, it can be noticed from Figure 10 and Figure A6 that the proportion of crystallized matter also has an influence on the drug release profile, however, to a lesser extent than the particle size of the composites.

## 4. Discussion

Until now, the MAL method was developed by milling APIs and unmodified MPS carriers systematically performed at 30 Hz. It was shown that it was a green (solvent free) loading method with high filling capacity. Using this method makes it possible to select the requested drug dosage up to about 40 wt %. It was also shown that hydrophobic materials can be loaded in large proportions. Additionally, a solid-state loading method provides the unique opportunity to load IBP in the nanocrystalline form coexisting with amorphous IBP within MPS carriers with an average pore diameter of 9.4 nm [14]. Below this size, IBP confined by MAL at 30 Hz was systematically found in the amorphous state. TEM analyses [14] have shown that the 2D ordered structure of channels was preserved after a milling time of 30 min at 30 Hz; only a reduction in the particle size was observed.

The present study has firstly focused on the analysis of the influence of the milling frequency on the particle size of SBA-15 and on the inherent consequences on the physical state and release profiles of confined IBP. SEM investigations revealed a change in the particle size distribution for milling frequencies from and below 20 Hz. For low milling frequencies, a bimodal size distribution was observed reflecting the presence of big particles not existing at 30 Hz. A direct relationship was determined between the particle size distribution and the physical state of IBP. For milling at 10 Hz, corresponding to a large population of big particles, more than 40% of IBP is crystallized. It is worth noting that a difference in I_QES_(T), especially a shift in the T_g_ of amorphous IBP, can be observed (in Figure 6b) after melting nanocrystals between IBP35%:SBA-15 composites prepared at 10 Hz and 30 Hz. This feature indicates a change in the temperature dependence of the viscosity of IBP; depending on the particle size, the viscosity increases with the particle size increase.

The study secondly aimed to determine the influence of the chemical modification of the pore surface, both on the physical state of the confined API and on the drug release profile, for composites prepared by the MAL method. It was found that the MAL method makes it possible to load a significant proportion of crystalline IBP in SBA-15-NH_2_—even by milling at 30 Hz almost 70% of the confined IBP is crystallized. For loading IBP at 10 Hz within SBA-15-NH_2_, the I_QES_(T) curve (X symbols) in Figure 7b clearly reflects the bimodal distribution of the particle size, with a first increase close to 50 °C corresponding to the melting of the smallest particles, and a stronger increase slightly above 70 °C corresponding to the melting of the biggest particles. The very low value of I_QES_ at room temperature confirms the high proportion of nanocrystalline IBP. Interestingly, it was shown that the particle size has a more significant influence on the drug release profile than the physical state, since profiles obtained for IBP35%:SBA-15 and IBP35%:SBA-15-NH_2_ prepared by milling at 10 Hz are almost superimposed and are different from that obtained for IBP35%:SBA-15-NH_2_ prepared at 30 Hz. It should be outlined that these findings can be only obtained using a solid-state loading method since crystallization within these types of MPS is not possible from the liquid state by using solvent-based loading methods.

## 5. Conclusions

This study shows that the MAL method provides the opportunity to tune a variety of operating parameters for administering customized drug-dosage forms. It was shown that tuning the milling intensity and modifying the chemical surface of pores to a lesser extent makes it possible to finely control the drug release profile. The next step will consist of deeply considering the influence of other architectural parameters of mesoporous silica carriers, including tortuosity and pore size, to broaden the spectrum of drug release profiles.

## Figures and Tables

**Figure 1 pharmaceutics-15-00390-f001:**
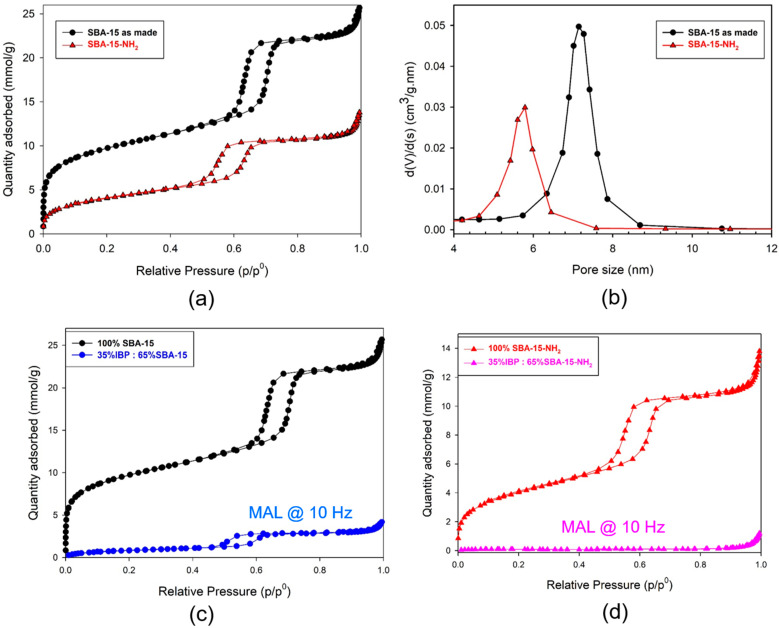
Characterization of mesoporous silica; (**a**) N_2_ adsorption isotherms for SBA-15 and SBA-15-NH_2_; (**b**) pore size distribution of SBA-15-NH_2_ as given by the BJH method; (**c**) comparison between N_2_ adsorption isotherms for SBA-15 and composites loaded by MAL at 10 Hz; (**d**) comparison between N_2_ adsorption isotherms for SBA-15-NH_2_ and composites loaded by MAL at 10 Hz.

**Figure 2 pharmaceutics-15-00390-f002:**
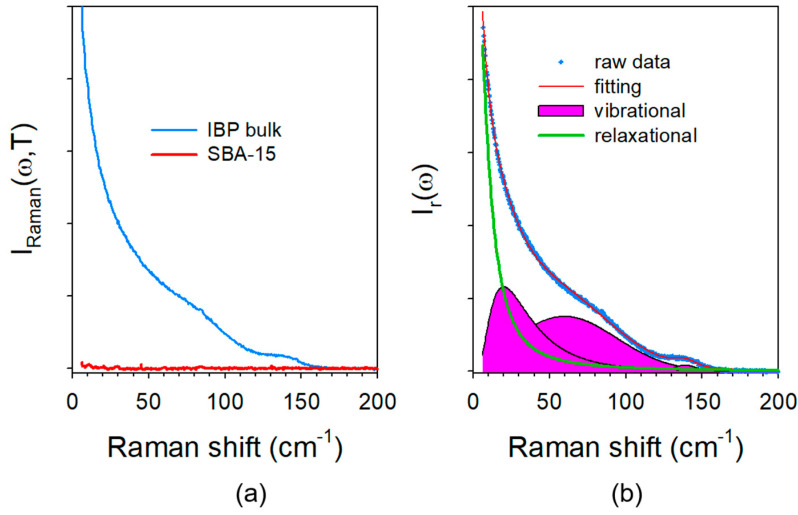
Low-frequency spectrum of IBP at room temperature; (**a**) Raman intensity of the undercooled liquid bulk of IBP compared with the spectrum of SBA-15; (**b**) fitting procedure of the reduced intensity used for discriminating the contributions of relaxational and vibrational motions.

**Figure 3 pharmaceutics-15-00390-f003:**
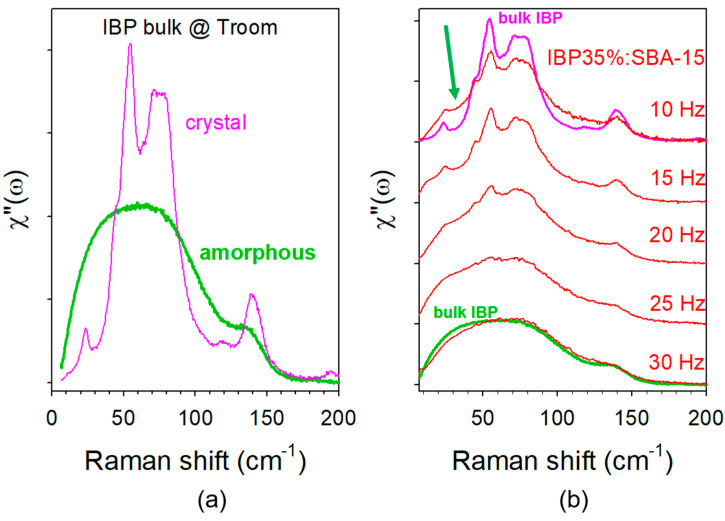
Raman susceptibility spectra of crystalline, partially crystalline, and amorphous ibuprofen; (**a**) spectra of the crystalline and amorphous bulk forms; (**b**) spectra of confined ibuprofen in SBA-15 using MAL at various frequencies: the arrow highlights the intensity excess in the very low-frequency range, reflecting the presence of amorphous IBP.

**Figure 4 pharmaceutics-15-00390-f004:**
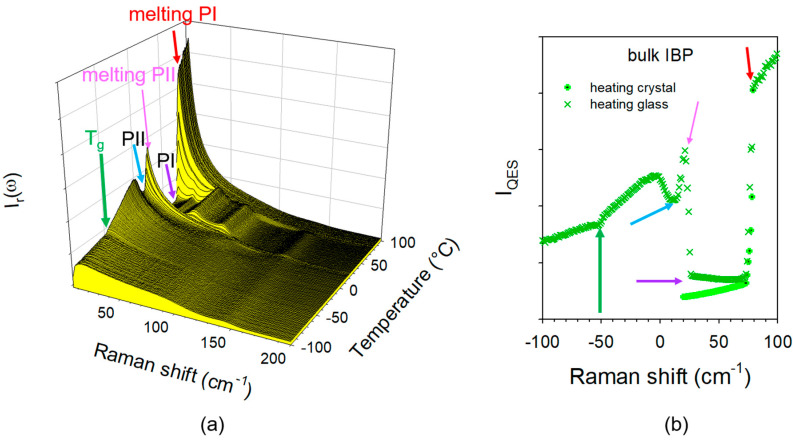
Analysis of the low-frequency spectrum of IBP bulk; (**a**) spectra collected during the heating ramp at 1 °C/min from the glassy state at −100 °C plotted in reduced intensity; (**b**) temperature dependence of the quasielastic intensity obtained by integrating the reduced intensity in the very low-frequency region. The arrows highlight the successive phase transitions undergone by IBP, i.e., the glass transition, crystallization in the metastable phase II (PII), the melting of PII, the crystallization of the stable phase I (PI), and the melting of PI.

**Figure 5 pharmaceutics-15-00390-f005:**
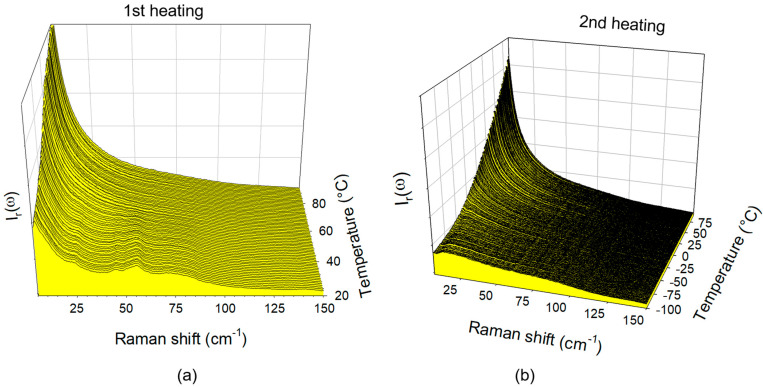
Low-frequency Raman spectra of the composite (IBP35%:SBA-15) prepared by MAL at 10 Hz collected upon heating at 1 °C/min; (**a**) first heating from room temperature up to 100 °C of the composite freshly prepared; (**b**) second heating of the composite from −100 °C.

**Figure 6 pharmaceutics-15-00390-f006:**
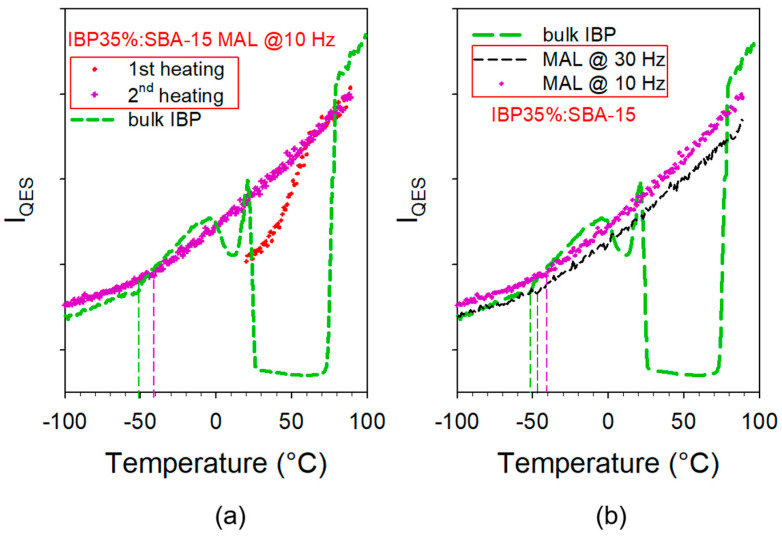
Temperature dependence of the quasielastic intensity (I_QES_) of the composite (IBP35%:SBA-15) compared with that of bulk IBP; (**a**) I_QES_ calculated in the first and second heating runs of the composite prepared by MAL at 10 Hz; (**b**) comparison between composites prepared by MAL at 10 Hz and 30 Hz.

**Figure 7 pharmaceutics-15-00390-f007:**
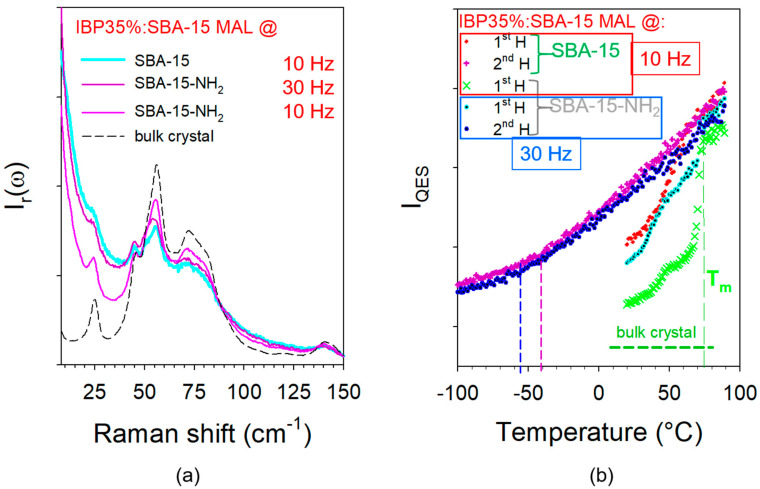
Low-frequency analysis of composites prepared with as-received (SBA-15) and modified (SBA-15-NH_2_) matrices by MAL at various frequencies; (**a**) spectra of freshly prepared composites taken at room temperature compared with the spectrum of the commercial phase; (**b**) temperature dependence of the quasielastic intensity (I_QES_) obtained upon a first heating from room temperature and a second successive heating; vertical dashed lines roughly indicate T_g_.

**Figure 8 pharmaceutics-15-00390-f008:**
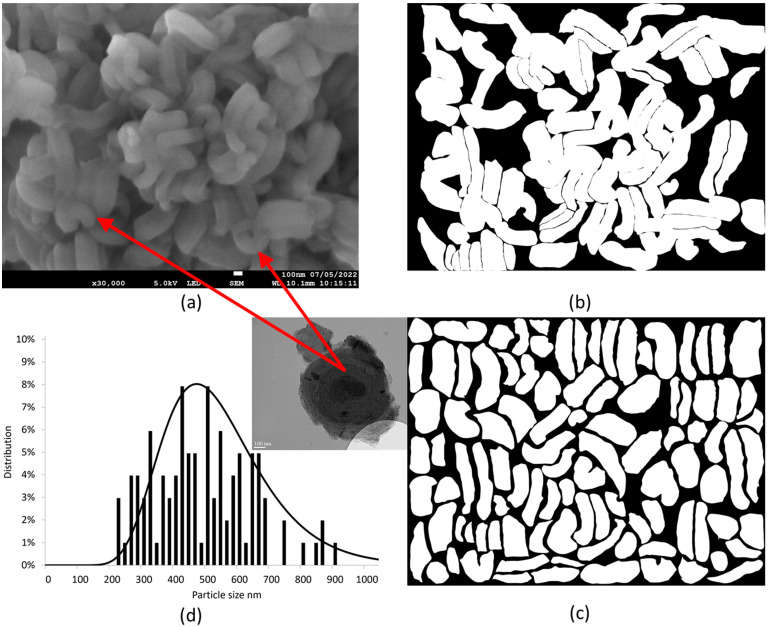
Methodology to obtain the distribution of particle sizes (example of an un-milled SBA-15 powder sample): (**a**) raw SEM micrography; (**b**) corresponding particles traced with hand; (**c**) isolation of particles; (**d**) corresponding distribution of particle sizes. Red arrows show local observations from TEM experiments, in various regions of the SEM micrography.

**Figure 9 pharmaceutics-15-00390-f009:**
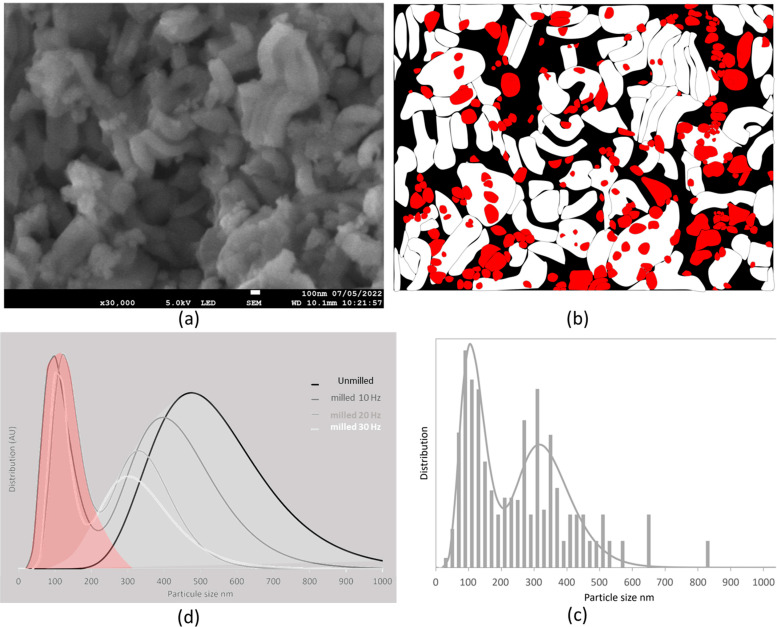
Distribution of particle sizes: (**a**) example of a raw SEM micrography of an SBA-15 specimen milled at 10 Hz; (**b**) corresponding particles traced with hand (small particles are colored in red and big particles are colored in white); (**c**) bimodal distribution of particle sizes of an SBA-15 specimen milled at 20 Hz; (**d**) evolution of the distribution of particle sizes with the milling frequency.

**Figure 10 pharmaceutics-15-00390-f010:**
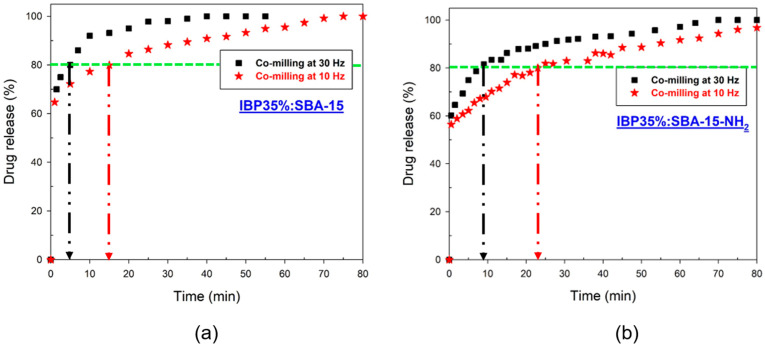
Release profiles of IBP (**a**) from SBA-15; (**b**) from SBA-15-NH_2_.

**Table 1 pharmaceutics-15-00390-t001:** Degree of crystallinity (%) of IBP confined in unmodified (SBA-15) and modified (SBA-15-NH_2_) carriers for various milling frequencies.

	30 Hz	25 Hz	20 Hz	15 Hz	10 Hz
SBA-15	0	4.5	19.6	39.2	42.2
SBA-15-NH_2_	67.8	--	--	--	77.8

**Table 2 pharmaceutics-15-00390-t002:** Evolution of particle and proportion size with milling frequency.

Milling Frequency (Hz)	Small Particle Sizes (nm)	Big Particle Sizes (nm)	Proportion of Small Particles (%)	Proportion of Big Particles (%)
Un-milled	--	470 ± 165	0	100
10	95 ± 50	405 ± 135	20	80
20	125 ± 45	335 ± 100	50	50
30	110 ± 30	305 ± 115	65	35


## Data Availability

Data related to this study are presented within this article.

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
