# Peer review of "Milling-Assisted Loading of Drugs into Mesoporous Silica Carriers: A Green and Simple Method for Obtaining Tunable Customized Drug Delivery"

_pharmaceutics, 2023, doi:10.3390/pharmaceutics15020390_

Round 1

Reviewer 1 Report

In paper entitled  Milling-Assisted Loading drugs into mesoporous silica carriers:  a green and innovative method for making easily and finely tunable the customized drug dosage and delivery” submitted by Hédoux and coworkers authors describe the solvent-free method of encapsulation of ibuprofen to SBA-15. This mechanochemical approach seems to be very attractive and can be considered an ecologically and economically viable alternative to the commonly used wet methods of loading guest particles into the pores of silica materials. The MAL (Milling-Assisted Loading) approach has also been tested by other groups using different mesoporous silica carriers.  The high applicability and usability of MAL has been confirmed.

In the work under evaluation, the authors chose Raman spectroscopy as the basic tool for studying the loading process. Raman spectroscopy is a field of expertise for Prof. Hédoux and his group, so it is no surprise that this part of the project is performed at a high level. My opinion on this article is positive, but a few points need to be clarified and described in more detail.

1) Very recently, Tricker et al. (Andrew W. Tricker,  George Samaras, Karoline L. Hebisch, Matthew J. Realff, Carsten Sievers Hot spot generation, reactivity, and decay in mechanochemical reactors Chemical Engineering Journal, 382, 2020, 122954) have presented theoretical approach to describe the reactive conditions in mechanochemical reactors.  They showed that during balls collisions in the ball-mills the hot spots with temperatures exceeding 1000 K that persist for tens of milliseconds can occurs. Such high temperature causes the local melting of API (in particular for low melting API as in case Ibuprofen, mp =78 deg C) what justify the question about common aspects between ball-milling and melting methods. In manuscript authors stated “To avoid any overheating of IBP, milling periods (5 min) were alternated with pause periods (2 min)” but this may not be enough. Authors should clearly explain which mechanism (MAL or melting) dominates. Perhaps both mechanisms cooperate in this case.

2) It is well known that during solvent-free loading methods, guest molecules can be located in different domains of mesoporous silica. These particles can be trapped inside the pores and/or located on the outer walls. In the course of the study, the authors demonstrated the effect of grinding frequency on the particle size of SBA-15. ”A direct relationship was determined between the particle 426 size distribution and the physical state of IBP. For milling at 10 Hz, corresponding to a  large population of big particles, more than 40 % of IBP is crystallized.” Probably, the alternative explanation can be also true. If IBP is used in crystalline form (commercial sample), then during grinding at 10 Hz IBP is outside the pores and retains its crystallinity. The problem of "inside/outside" locating the guest molecule in the SBA should be thoroughly explained.

3) Referring to the discussion in point 2, the interpretation of the release profile also requires additional comment.  The authors stated that the IBP release is “”two steps process, (i) a very fast release observed up to 5min, most probably due to IBP molecules located in the core of the channels, followed by (ii) a slow release attributed to the IBP  molecules, which interact with the surface of the pores” . According to the presented results, the sample is fully amorphized when the milling frequency is 30 Hz. Can the authors say what is the proportion of IBP inside the channel compared to IBP on the surface? What profile can we expect when we have an amorphous phase inside/outside the pores and a significant contribution of crystalline phase?

4) In all experiments, the grinding time is 30 minutes. Isn't it too long for a milling frequency of 30 Hz. Did the authors try to optimize the grinding time?

5) DSC studies should support measurements of variable temperature Raman spectroscopy. I encourage authors to do such research.

Reviewer 2 Report

In their work “Milling-assisted loading drugd inot mesoporous silica carriers […]” Moutamenni co-authors propose a solvent free method based on co-milling for loading drugs in mesoporous silica. Although this technique is not new, the novelty of the paper resides in the study of the influence of milling condition (i.e. milling frequency) and of the chemical modification of the support surface on the crystallinity of the loaded drug and the consequent release behavior. The study is carried put on SBA-15 as a support and ibuprofen as a model drug. On the overall, the paper is interesting and gives new insights in the co-milling loading of drug, however it needs some improvements. In particular, here are my comments/suggestions.

-        Title: I find the title very long and not clear. I suggest something like: “Milling-assisted loading of drugs into mesoporous silica: a green and simple method for obtaining tunable customized drug delivery”.

-        Lines 44-48: this sentence is not clear to me. If the pore size is larger than the critical size of nucleation does the nucleation occur or not? Please rephrase it.

-        Line 84: could you explain why is amorphization by milling impossible if the temperature is higher that Tg?

-        Line 98: please add details on the type of SBA-15 used (hexagonal symmetry, pore diameter XX nm…).

-        Line 133: erase the sentence “Figure 8d shows an example of an un-milled SBA specimen”. This should be said in the results section and not in the materials and methods one.

-        Line 139-141: where sample for SEM metallized?

-        Line 144: replace “performed” with “monitored”

-        Line 146: “…maximum of absorbance of ibuprofen in 0.1 M HCl solution”. I would add this last part since the peak of absorbance varies depending on the solvent.

-        Line 149: “…0.1 M hydrochloric acid solution…”

-        Drug release studies: how long was the test? What was the sampling frequency? Please add these details.

-        Line 183: erase “and is used for obtaining structural information in molecular disordered systems” because this is a repetition of line 176.

-        Line 189: replace “Erreur ! Source du renvoi introuvable” with “Fig.1.a”. Check the entire manuscript, because this error is present in several places.

-        Line 382-383: when I saw the release curves I thought that the fast initial release was due to IBP that is located on the external surface of silica particles rather than it its porosities. Why do you exclude this possibility?

-        Linked to the previous point: I encourage you to provide N2 adsorption isotherms of the material after IBP was loaded. These data, compared to those presented in fig. 1, can provide fundamental hints about the location of IBP. (this, in my opinion, is a major revision that should be done)

-        Figure 10: I guess that the reported percentage of drug release is the percentage compared to the final amount released. This means that the percentage was calculated dividing the released amount at every t by the released amount at 80 min. Am I right? If it is so, I wonder what is the absolute amount of IBP released. Does it corresponds to the total loaded amount? That is to say, is IBP totally released?

-        Line 404-405: the strong influence of the size distribution of silica particles could be coherent with my hypothesis of IBP location on the external surface of the particle, since smaller particles have a higher specific surface.

-        Line 415-416: what is the advantage of having drugs in nanocrystals? As far as I know, the amorphous form is better because, being less stable, it favors drugs solubility.

-        Line 417: where come this average pore diameter (9.4 nm) from? This was never mentioned before.

   Please, check English throughout the manuscript, because there are several flaws. Here are some:

-        line 37: replace “improving” with “improve”

-        line 62: …30 Hz makes it possible to…

-        line 68-69: …making the crystallization in very small pore diameters possible…

-        line 81: …without changing…

-        Line 191: erase “For instance” at the beginning of the sentence

-        Caption of fig.1: 2 must be a subscript in N2 and NH2.

-        Caption of fig.2: “spectrum” instead of “spectru”

-        Line 416: replace “nanocrystal state” with “nanocrystal form”. Crystallinity is not a state of the matter.

Round 2

Reviewer 1 Report

I accept the explanations and corrections made by the authors in the revised version.

Reviewer 2 Report

I thank the authors for the extensive answers and the modifications of the manuscript.